# Synergism of Carbamoylated Erythropoietin and Insulin-like Growth Factor-1 in Immediate Early Gene Expression

**DOI:** 10.3390/life13091826

**Published:** 2023-08-29

**Authors:** Morgan J. Rothschadl, Monica Sathyanesan, Samuel S. Newton

**Affiliations:** Division of Basic Biomedical Sciences, Sanford School of Medicine, University of South Dakota, Vermillion, SD 57069, USA; morgan.j.rothschadl@coyotes.usd.edu (M.J.R.); monica.sathyanesan@usd.edu (M.S.)

**Keywords:** gene regulation, PC-12, neurotrophic signaling, cognition, combination therapies

## Abstract

Trophic factors are secreted proteins that can modulate neuronal integrity, structure, and function. Previous preclinical studies have shown synergistic effects on decreasing apoptosis and improving behavioral performance after stroke when combining two such trophic factors, erythropoietin (EPO) and insulin-like growth factor-1 (IGF-1). However, EPO can elevate the hematocrit level, which can be life-threatening for non-anemic individuals. A chemically engineered derivative of EPO, carbamoylated EPO (CEPO), does not impact hematological parameters but retains neurotrophic effects similar to EPO. To obtain insight into CEPO and IGF-1 combination signaling, we examined immediate early gene (IEG) expression after treatment with CEPO, IGF-1, or CEPO + IGF-1 in rat pheochromocytoma (PC-12) cells and found that combining CEPO and IGF-1 produced a synergistic increase in IEG expression. An in vivo increase in the protein expression of Npas4 and Nptx2 was also observed in the rat hippocampus. We also examined which kinase signaling pathways might be mediating these effects and found that while AKT inhibition did not alter the pattern of IEG expression, both ERK and JAK2 inhibition significantly decreased IEG expression. These results begin to define the molecular effects of combining CEPO and IGF-1 and indicate the potential for these trophic factors to produce positive, synergistic effects.

## 1. Introduction

Trophic factors are secreted proteins that have well-defined roles in modulating neuronal integrity, structure, and function. These roles can range from cell survival to neuronal differentiation to neurite outgrowth, with an individual trophic factor generally having unique, multimodal effects [1,2]. The level at which these neurotrophic factors are expressed in particular brain regions plays a major role in facilitating these multimodal functions, with decreased levels of neurotrophic factors generally correlating with synaptic injury and decreases in cognitive functioning. Wang et al. (2020) noted a positive correlation between decreases in brain-derived neurotrophic factor (BDNF) in the serum and cognitive impairments in a patient model of panic disorder [3]. A separate meta-analysis of fifteen studies found that Alzheimer’s disease (AD) patients had decreased levels of serum BDNF compared to healthy controls [4]. While BDNF has been the primary focus, it is important to note that erythropoietin (EPO) and insulin-like growth factor-1 (IGF-1) have been reported to exhibit robust cognitive effects in both preclinical [5,6,7] and clinical studies [8,9,10,11,12].

While both EPO and IGF-1 have shown positive, pro-cognitive benefits in multiple studies when used individually, some studies have shown a synergistic increase in effectiveness when testing these trophic factors in combination. Digicaylioglu et al. (2004) and Fletcher et al. (2009) investigated neuroprotection after treatment with a combination of EPO and IGF-1 and found that the combination approach provided more robust protection from injury compared to when either trophic factor was used independently [13,14]. As well, Kang et al. (2010) tested EPO and IGF-1 in combination in HIV-associated neurocognitive disorders (HAND) and found a striking decrease in hyperphosphorylated tau, one proposed driver of cognitive decline in both HAND and AD, after combination treatment compared to vehicle administration [15]. Despite the effectiveness seen by this combination, the use of EPO as a pro-cognitive treatment is limited due to its role in elevating red blood cell counts, which can be life-threatening for non-anemic individuals. This limitation has been addressed by the development of non-erythropoietic EPO derivatives such as carbamoylated erythropoietin (CEPO) [16]. CEPO is a chemically engineered EPO derivative that has shown both neurotrophic and pro-cognitive benefits similar to those seen by EPO, but with no effect on the hematological parameters [17,18].

That these trophic factors can promote neuroprotective and pro-cognitive benefits is well documented, but the mechanism underlying these effects is unknown. One hypothesis is that these trophic factors could promote synaptic rearrangement, which is important for memory formation and overall synaptic plasticity [19,20,21]. In the brain, immediate early gene (IEG) upregulation signals the start of the synaptic rearrangement process, as well as being a marker of neuronal activity. In response to external stimuli, these genes are rapidly and transiently upregulated as they do not require new proteins to first be synthesized. There are two main subsets of IEGs, those that are rapidly induced (within 5–30 min) and those that are slowly induced (within 1–6 h). Generally, the rapidly induced IEGs encode for transcription factors while the more slowly induced IEGs encode for effector proteins [22].

While IEGs are present in almost all cell types, their upregulation in the brain can produce neuronal-specific outcomes [23]. These outcomes can range from impacting memory consolidation to maintaining the excitatory/inhibitory balance of neurons to functioning as transcription factors and changing the expression of downstream genes [22,24]. These IEG-specific neuroplastic changes work together to begin a cascade of events that result in new proteins being synthesized, which is a crucial step in the consolidation of a memory. We hypothesize that trophic factors produce cognitive effects, in part, by inducing IEG expression. This, in turn, will enhance the downstream effects, such as new protein synthesis and synaptic remodeling, which will lead to improved memory consolidation and pro-cognitive effects.

In the present study, we hypothesized that CEPO and IGF-1, when used in combination, would synergistically activate IEGs. To investigate this, we exposed PC-12 cells to either CEPO, IGF-1, or a combination of CEPO + IGF-1 at multiple time points and concentrations. We investigated which signaling pathways might be mediating these effects and whether the individual trophic factors activated different pathways. These results indicate that trophic factors can impact IEG expression, with a combination approach of CEPO and IGF-1 being more effective in upregulating this expression than when used independently.

## 2. Materials and Methods

### 2.1. Carbamoylation of EPO

Erythropoietin was purchased from Prospec Bio (Rehovot, Israel) and carbamoylated in 1 mg aliquots as mentioned previously [25]. Briefly, EPO was deprotonated in a high pH borate buffer (pH = 8.9) and then exposed to potassium cyanate for 16 h at 36 °C. CEPO was exhaustively dialyzed for 6 h against PBS. CEPO concentration was determined using the Nanodrop 2000 Spectrophotometer (ThermoFischer, Waltham, MA, USA).

### 2.2. Cell Culture

Rat pheochromocytoma (PC-12) cells were obtained from American Type Culture Collection (ATCC, Manassas, VA, USA). The cells were grown in suspension in RPMI-1640 (ATCC) with 10% heat-inactivated horse serum (Gibco, Grand Island, NY, USA) and 5% fetal bovine serum (Gibco) at 37 °C and 5% CO_2_. To differentiate the cells into neuronal cells [26], 5 × 10^4^ cells were grown for 10 days on collagen-coated dishes (Corning, Corning, NY, USA) in RPMI-1640 with 1% heat-inactivated horse serum and nerve growth factor (NGF) (100 ng/mL, Alomone Labs, Jerusalem, Israel). The medium was changed every two days to fresh differentiation medium. NGF was removed overnight before the day of the experiment. Neuronal morphology and neurite outgrowth were assessed via microscopy before proceeding.

### 2.3. Trophic Factor Treatment

After differentiating the PC-12 cells as described above, cells were treated for either one or three hours with CEPO (100 ng/mL), IGF-1 (100 ng/mL), CEPO + IGF-1 (25 ng/mL each; 1 h only), CEPO + IGF-1 (100 ng/mL; 3 h only), or CEPO + IGF-1 (50 ng/mL each; both 1 and 3 h) at 37 °C and 5% CO_2_ in RPMI-1640 with 1% heat-inactivated horse serum. Vehicle-treated (PBS) cells were used as controls. After the allotted time, RNA or protein were extracted using the RNAqueous Total RNA Isolation Kit (ThermoFischer) or RIPA buffer supplemented with 10 µL/mL phosphatase and protease inhibitors (ThermoFischer), respectively.

### 2.4. Inhibitor Studies

After differentiating the PC-12 cells as described previously, cells were incubated in RPMI-1640 with 1% heat-inactivated horse serum and one kinase inhibitor (MK-2206, 5 µM, 1 h; GDC-0994, 10 µM, 1 h; AZD1480, 2 µM, 2 h; Selleckchem, Houston, TX, USA) at 37 °C with 5% CO_2_. Cells were then stimulated with either CEPO (100 ng/mL), IGF-1 (100 ng/mL), or CEPO + IGF-1 (50 ng/mL each) in the presence of inhibitor for an additional hour at 37 °C with 5% CO_2_ before extracting RNA using the RNAqueous Total RNA Isolation Kit (ThermoFischer). Vehicle-treated (PBS) cells treated with inhibitors were used as the control.

### 2.5. cDNA Preparation

RNA was quantified using the Nanodrop 2000 spectrophotometer (ThermoFischer). RNA quality was assessed using the Nano kit on the Bioanalyzer (Agilent, Santa Clara, CA, USA). All RNA integrity numbers were ≥9. For each sample, 500 ng of RNA was amplified and reverse transcribed. The reaction was stopped by the addition of 0.5 M NaOH/50 mM EDTA before denaturing the RNA/DNA hybrids at 65 °C. This reaction was then neutralized by adding 1 M Tris-HCl, pH 7.5, before adding 10 mM Tris/1 mM EDTA, pH 8.0. Ethanol precipitation of the cDNA was performed utilizing linear acrylamide as the co-precipitant before adding 5 M NaCl and 100% Ethanol to the tubes and incubating the samples at −20 °C overnight. The next day, the cDNA mixture was spun at 10,000 rpm for 15 min at 4 °C. The alcohol/salt mix was mostly removed, leaving approximately 50 µL in the tube before adding cold 70% ethanol to the tubes. This mixture was again spun at 10,000 rpm for 15 min at 4 °C before removing most of the supernatant, leaving approximately 50 µL in the tube. The cDNA was then dried at 65 °C before reconstituting in 100 µL nuclease-free water.

### 2.6. Quantitative PCR

Quantitative PCR (qPCR) analysis was used to assess relative gene expression when compared to the vehicle-treated samples using the ΔΔCt method. Briefly, reverse transcribed cDNA was amplified employing SYBR Green chemistry (Qiagen, Germantown, MD, USA) and gene-specific primers (Appendix A) in the Mastercycler realplex Real-time PCR machine (Eppendorf, Hamburg, Germany). The PCR conditions were as follows (40 cycles): Denaturation—94 °C for 2 s, Annealing—60 °C for 30 s, Extension—72 °C for 30 s. Specificity of product was determined by melt curve analysis. Data were normalized using the housekeeping gene cyclophilin.

### 2.7. Western Blot Analysis

Equivalent amounts of protein (100 µg) were resolved on SDS-PAGE gels (7.5% resolving and 4% loading) under reducing conditions and transferred onto a polyvinylidene difluoride membrane (Bio-Rad, Hercules, CA, USA). After blocking in Intercept TBS Blocking Buffer (Li-Cor, Lincoln, NE, USA), the membranes were incubated with 0.2% Tween-20 and primary antibodies overnight at 4 °C followed by 0.2% Tween-20, 0.01% Sodium Dodecyl Sulfate, and either IRDye 800CW or IRDye 680RD (Li-Cor) for one hour at room temperature. Protein bands were imaged using the LiCor Odyssey CLx. Band intensities were quantified using ImageJ. The following primary antibodies were used: p-AKT (Ser-473, Cell Signaling Technologies, Danvers, MA, USA), AKT (Cell Signaling Technologies), p-ERK (Thr202/Tyr204, Cell Signaling Technologies), ERK (Cell Signaling Technologies), and β-actin (Santa Cruz Biotechnology, Santa Cruz, CA, USA).

### 2.8. Digital Droplet PCR

To compare the number of target DNA per cell, digital droplet PCR (dPCR) was utilized using the QIAcuity One Digital PCR System. Samples were partitioned using the 8.5 K nanoplates (Qiagen) and PCR amplified utilizing gene specific primers (Appendix A) and EvaGreen technology (Qiagen). For the calculations, first the reaction volume was divided by the sample volume and then multiplied by the standard concentration readout given by the instrument. This value was then multiplied by the sample volume and any dilutions from the original sample were taken into account to determine the copies of target DNA per reaction. The number of target DNA per cell was then calculated by taking the average concentration of RNA and multiplying that by the total volume of RNA captured after extraction. The number of cells used in culture were then divided by the nanograms of RNA calculated previously and then multiplied by the total amount of RNA used to make cDNA to calculate the total amount of cells used in our cDNA preparation. Finally, the copies of target DNA per reaction (calculated previously) were divided by the total amount of cells to determine the copies of target DNA per cell.

### 2.9. Animal Studies

Adult male Sprague Dawley rats (*n* = 5 vehicle, *n * = 6 CEPO + IGF-1; mass 222–290 g) were maintained on a standard 12 h light–dark cycle with free access to food and water for the duration of the experiment. All procedures were carried out in strict accordance with the National Institutes of Health Guide for the Care and Use of Laboratory Animals and approved by the USD Institutional Animal Care and Use Committee on 13 February 2023, under protocol number 01-01-23-26D. Every effort was made to minimize the number of animals used. Rats received singly daily i.p. injections of either vehicle (PBS) or CEPO (25 µg/kg) + IGF-1 (50 µg/kg) for four consecutive days. These doses were chosen based on previous in vivo work, with a slight decrease in CEPO dosing to be half of the IGF-1 dose [25,27]. Five hours after the final dose, animals were decapitated according to American Veterinary Medical Association guidelines and the brains were frozen on dry ice.

### 2.10. Immunohistochemistry

Immunohistochemical studies were performed using cryocut hippocampal sections (coronal, 16 µm) as previously described [28], with slight changes in blocking. Sections were blocked with either normal chicken serum (Nptx2, Vector S3000, Vector Laboratories, Burlingame, CA, USA) or normal goal serum (Npas4, Vector S1000, Vector Laboratories) for one hour at room temperature. Sections were incubated with different primary antibody concentrations in antibody solution (2.5% BSA in PBS) at 4 °C overnight (Nptx2, 1:100, Abcam, Cambridge, England; Npas4, 1:200, Abcam). Following primary antibody incubation, slides were washed in 1xPBS three times for five minutes each at room temperature. Slides were then incubated with appropriate fluorescent secondary antibody (1:500, Alexa-488 and 594, Invitrogen, Carlsbad, CA, USA) in antibody solution for 1.5 h at room temperature. The slides were then rinsed in 1× PBS three times for five minutes each and cover slipped using VectaMount (Vector Laboratories). Sections were viewed and images were captured using a Nikon Eclipse Ni microscope equipped with DS-Qi1 monochrome, cooled digital camera and NIS-AR 4.20 Elements imaging software. CEPO + IGF-1 and vehicle-treated sections were captured using identical exposure sections. Sections = −3.30 mm from Bregma.

### 2.11. Statistical Analysis

All statistical analyses were performed using GraphPad Prism 8.4.3. Outliers greater than 2 standard deviations away from the mean were removed from further analysis. Normality and lognormality were assessed using the Shapiro–Wilk test. If lognormality was needed, data were transformed before being analyzed for significance. A one-way ANOVA with Dunnett’s multiple comparisons post hoc test was used to look for significance in the qPCR studies when analyzing each gene individually and in the Western blot studies when analyzing phosphoprotein/total protein values. When neither normality nor lognormality were achieved, data were analyzed non-parametrically using the Kruskal–Wallis test with Dunn’s multiple comparisons post hoc test.

## 3. Results

### 3.1. CEPO + IGF-1 Combination Therapy Robustly Activates IEG Expression

IEGs in the brain have long been a marker for neuronal activity, with their induction as a response to a behavioral stimulus. To determine whether these IEGs are upregulated in response to trophic factor treatment, PC-12 cells were treated with CEPO (100 ng/mL), IGF-1 (100 ng/mL), and CEPO + IGF-1 (50 ng/mL each) for either one (*n* = 6/treatment group) or three (*n* = 5/treatment group) hours. After one hour of exposure, robust upregulation of *Fos proto-oncogene* (*cFos*), *FosB proto-oncogene* (*FosB*), *JunB proto-oncogene* (*JunB*), *early growth response 1* (*Egr1*), and *activity-regulated cytoskeleton-associated protein* (*ARC*) were noted, with the combination approach (50 ng/mL of each trophic factor) showing the highest increase in expression of these five genes. IGF-1 alone also showed an upregulation in these five genes while CEPO alone did not impact the expression of any of these genes. IGF-1 alone and combination treatment (50 ng/mL each), but not CEPO alone, also showed upregulation in *brain-derived neurotrophic factor* (*BDNF*) expression after one hour. CEPO increased *neuronal PAS domain protein 4 (Npas4*), *inhibin subunit beta A* (*Inhba*), and *neuronal pentraxin-2 precursor (Nptx2*) expression after one hour of treatment while IGF-1 increased expression of *Npas4* and *Nptx2*, but not *Inhba*. The combination approach did not significantly alter gene expression for *Inhba* or *Nptx2* but did for *Npas4*. A separate, low-dose combination approach using 25 ng/mL of each trophic factor (*n* = 6/treatment group) was also tested at one hour. While significant upregulation was seen for *cFos*, *FosB*, *JunB*, *Egr1*, *Npas4*, and *ARC*, the upregulation was not as robust as seen in the combination approach where 50 ng/mL of each trophic factor was used (Figure 1A).

After three hours of exposure, the increase in gene expression differentially altered. After both IGF-1 alone and combination treatment (50 ng/mL each), *JunB*, *Egr1*, and *ARC* still showed significant upregulation, while only showing changes after CEPO treatment for *Egr1*. *FosB* and *cFos* no longer showed any significant changes after IGF-1 or CEPO + IGF-1 (50 ng/mL each) treatments. Combination treatment (50 ng/mL each) also showed upregulation in *BDNF* expression while both CEPO alone and IGF-1 alone trended towards a non-significant increase in expression. CEPO treatment still significantly upregulated *Nptx2* expression, but no longer significantly altered *Npas4* or *Inhba* expression. Combination treatment (50 ng/mL each), however, did upregulate *Npas4*, *Inhba*, and *Nptx2* expression, which was not seen at the one-hour time point. For the three-hour time point, a separate combination approach using 100 ng/mL of each trophic factor (*n* = 4/treatment group) was also tested and showed significant gene expression changes for *cFos*, *JunB*, *Egr1*, and *ARC*, with the other genes showing little to no changes in expression after this high-dose combination treatment (Figure 1B). In general, this high dose (100 ng/mL each) combination approach resulted in a decrease in gene expression, giving a bell-shaped concentration versus expression curve. This bell-shaped dose–response curve has been reported previously for various cytokine and growth factor receptors, including both the erythropoietin receptor and the insulin-like growth factor-1 receptor [29,30,31,32,33]. Previous reports have indicated that in the presence of high ligand concentrations, each monomeric unit of the dimeric receptor becomes occupied with a ligand, which does not allow the receptor to dimerize and act as an agonist, inhibiting the receptor from signaling [29]. This could explain why we see a decrease in gene expression as the concentration of each trophic factor is doubled. In conjunction with these results, it appears that of the three combination treatments, the approach that utilizes 50 ng/mL of each trophic factor showed the most robust upregulation in gene expression. As well, CEPO and IGF-1 treatments appear to be acting synergistically during the combination treatments to further upregulate gene expression compared to the individual treatments.

### 3.2. IGF-1, but Not CEPO, Upregultes AKT and ERK Phosphorylation

To determine which kinase signaling pathways are being impacted by trophic factor stimulation, PC-12 cells were exposed to CEPO (100 ng/mL), IGF-1 (100 ng/mL), or a combination of CEPO + IGF-1 (50 ng/mL each) for either five minutes, one hour, three hours, or five hours. After all four time points, exposure to CEPO showed no changes in AKT (Ser473) phosphorylation while exposure to either IGF-1 or both CEPO + IGF-1 showed robust increases in expression (Figure 2B,E and Appendix A). Our results do not show a synergistic increase in AKT phosphorylation after combination CEPO + IGF-1 treatment in PC-12 cells. For ERK1/2 (Thr202 and Tyr204), robust increases in phosphorylation were seen after both five minutes and one hour of IGF-1 and CEPO + IGF-1, but not CEPO, stimulation (Figure 2C,F). ERK1 and ERK2 phosphorylation patterns differed slightly. For ERK1, both IGF-1 and CEPO + IGF-1 treatments showed similar increases in phosphorylation after five minutes and one hour of treatment. For ERK2, differences in expression were seen after treatment with IGF-1 or CEPO + IGF-1, with the five-minute treatment showing higher ERK2 expression after IGF-1 alone compared to CEPO + IGF-1 while the one-hour treatment showed higher ERK2 expression after CEPO + IGF-1 treatment compared to IGF-1 alone. After both three and five hours of stimulation, ERK1/2 phosphorylation decreased below vehicle-treated levels, showing significant decreases in phosphorylation after three hours of trophic factor stimulation and no decreases in phosphorylation after five hours of stimulation (Appendix A). These results indicate that IGF-1 and combination CEPO + IGF-1, but not CEPO, stimulation cause both a robust and a prolonged phosphorylation of both AKT and ERK1/2, with the AKT phosphorylation lasting longer than the ERK phosphorylation.

### 3.3. ERK and JAK2, but Not AKT, Inhibition Cause a Decrease in IEG Expression

To further elucidate which kinase signaling molecules might be mediating the upregulation in IEG expression, PC-12 cells were pretreated with either an AKT (MK-2206; 5 µM, 1 h), ERK (GDC-0994; 10 µM, 1 h), or JAK2 inhibitor (AZD1480; 2 µM, 2 h) before being stimulated with either CEPO (100 ng/mL), IGF-1 (100 ng/mL), or CEPO + IGF-1 (50 ng/mL each) in the presence of inhibitor for one hour. The vehicle-treated cells also received inhibitors similar to the trophic factor-treated cells. The expression changes in these graphs are being compared to the pattern of gene expression seen in Figure 1A. AKT inhibition showed few changes in the expression of *cFos*, *FosB*, *JunB*, *Egr1*, and *ARC*, with the main change being an increase in the overall expression of *Egr1* after both IGF-1 and CEPO + IGF-1 treatments (Figure 3C). A decrease in the expression of *Npas4*, *Inhba*, and *Nptx2* after CEPO treatment is noted after AKT inhibition as these no longer show significant upregulation in expression. *BDNF* no longer shows any significant changes in expression after either IGF-1 or CEPO + IGF-1 treatment when AKT is inhibited. Despite minimal changes after AKT inhibition, both ERK and JAK2 inhibition caused significant decreases in IEG expression.

After ERK inhibition, *cFos*, *JunB*, and *ARC* still showed significant changes in gene expression after IGF-1 treatment while *JunB*, *Egr1*, and *ARC* showed significant changes in expression after CEPO + IGF-1 exposure, albeit at much lower levels than seen previously when no inhibitor was present. *FosB* was also highly downregulated and no longer showed any significant changes in expression. *BDNF* still showed upregulation after IGF-1 treatment, but no longer showed significant upregulation after CEPO + IGF-1 treatment. After CEPO treatment, *Npas4*, *Inhba*, and *Nptx2* no longer showed any significant changes in expression (Figure 3A). The genes *cFos*, *FosB*, *JunB*, *Egr1*, and *ARC* also showed significant decreases in expression after JAK2 inhibition, with *cFos* showing significant upregulation after CEPO, IGF-1, and CEPO + IGF-1 exposure and *JunB* and *ARC* showing significant upregulation after IGF-1 and CEPO + IGF-1 exposure. *BDNF* showed significant upregulation after CEPO + IGF-1 treatment, but not after IGF-1 alone. While *Npas4*, *Inhba*, and *Nptx2* no longer showed any significant upregulation after CEPO treatment, they did show robust increases after combination treatment, which was not seen in the absence of inhibitor (Figure 3B).

### 3.4. IGF1R Levels Are More Than 50-Fold Higher Than EPOR, CD131, or IR in PC-12 Cells

To understand whether receptor levels played a role in the varying gene expression, dPCR was utilized to quantify the number of receptors in our PC-12 cells. Receptor levels were calculated per cell (Table 1). Our four receptors of interest were the *beta common receptor* (*CD131*), the *erythropoietin receptor* (*EPOR*), the *insulin-like growth factor-1 receptor* (*IGF1R*), and the *insulin receptor* (*IR*), as these are the receptors to which our trophic factors (CEPO and IGF-1) bind. When looking at the number of receptors/cell, the expression levels from lowest to highest were: *CD131*, *IR*, *EPOR*, *IGF1R*, with the number of *IGF1R* receptors/cell being over 50-fold higher than the number of receptors/cell for either *CD131*, *IR*, or *EPOR*.

The expression levels of these receptors were also analyzed via qPCR after both one and three hours of treatment with either CEPO (100 ng/mL), IGF-1 (100 ng/mL), or CEPO + IGF-1 (50 ng/mL each). After one hour of CEPO + IGF-1 stimulation, *IGF1R* showed significant upregulation while *IR* showed significant downregulation. *IR* also showed significant downregulation after CEPO treatment, but not after IGF-1 treatment (Figure 4A). After three hours of stimulation, *IGF1R* showed significant upregulation after CEPO treatment, but no longer after CEPO + IGF-1 treatment. *CD131* showed upregulation after CEPO + IGF-1 treatment (Figure 4B). *EPOR* showed no significant changes in expression after any of the treatments or time points.

### 3.5. Npas4 and Nptx2 Are Regulated by CEPO + IGF-1 in the Rat Hippocampus

To determine whether combining CEPO and IGF-1 was able to regulate IEG expression in vivo, we injected male rats with CEPO (25 µg/kg) + IGF-1 (50 µg/kg) for four days before observing hippocampal Npas4 and Nptx2 protein expression. An outline of the hippocampal regions shown in Figure 5 in the context of the entire hippocampus can be found in Appendix A. Both Nptx2 and Npas4 showed higher levels of protein expression after CEPO + IGF-1 treatment, albeit in different hippocampal regions. Nptx2 was regulated in the CA3 while Npas4 was regulated in the CA1 (Figure 5). These data provide validation that combining CEPO and IGF-1 can regulate IEGs at the protein level.

## 4. Discussion

Neurotrophic factors have long been associated with positive neuronal effects such as increasing cell survival and activating neurite outgrowth. Recent studies have started using combinations of trophic factors, namely EPO and IGF-1, both of which have known neurotrophic and neuroprotective properties when used individually. Determining whether these trophic factors can act synergistically, to potentially improve functional outcome in CNS diseases, is of significant interest. While these studies have shown promising in vitro and in vivo results, the use of EPO is limited due to its role in increasing red blood cell counts. However, similar studies utilizing EPO derivatives, such as CEPO, have not been done. Thus, it is not known whether these synergistic actions are translatable to other trophic factor combinations.

In this study, we used IEG expression to determine whether combining CEPO and IGF-1 could play a synergistic role in neurons. Functionally, IEGs are implicated in learning and memory mechanisms, with multiple IEGs being implicated in synaptic rearrangement, an important process in increasing cognitive capacity and one of the initial processes preceding memory formation. We used ten well-known IEGs, *cFos*, *FosB*, *JunB*, *Egr1*, *Npas4*, *Inhba*, *Arc*, *Nptx2*, *tPA*, and *BDNF*, to perform gene expression analyses after CEPO, IGF-1, or combination CEPO + IGF-1 treatment. The results from our study demonstrate that CEPO and IGF-1 synergistically regulated IEG expression in vitro in PC-12 cells.

The combination of CEPO and IGF-1 increased the protein expression of Nptx2 and Npas4 in the rat hippocampus. Neuronal pentraxin-2 (Nptx2), which is also commonly referred to as neuronal activity-regulated pentraxin, or Narp, and neuronal PAS domain protein 4 (Npas4) both play roles in the excitatory/inhibitory balance in the brain. Nptx2 forms a complex with other pentraxin proteins and helps to cluster α-amino-3-hydroxy-5-methyl-4-isoxazolepropionic acid (AMPA) receptors (AMPA-Rs) at the excitatory synapse [34,35,36]. Nptx2 has also recently been shown to help regulate complement, with an increase in the classical complement pathway and a subsequent decrease in excitatory synapses being seen after Nptx2 deletion [37]. As well, Nptx2 has been shown to be decreased in the post-mortem brains of people with AD [38] and in the cerebrospinal fluid of patients with schizophrenia [39]. Npas4 is a neuronal-specific IEG that alters the levels of inhibitory synapses on both cell bodies and apical dendrites [24,40]. A recent study implicated Npas4 in double-stranded break (DSB) repair after neuronal activity by forming a complex with NuA4 and targeting the damaged elements. This study not only showed that Npas4 disruption caused an increase in DSBs, but also showed that Npas4 disruption caused a decrease in lifespan for both male and female mice [41]. Npas4 deletion from the CA3 region of the hippocampus has also been shown to impair long-term contextual memory [42]. Since Npas4 is a transcription factor, it can impact the expression of downstream genes, with research showing Npas4-dependent regulation of *BDNF*, *Homer1a*, and *Nptx2* [27,36,37]. Our results showing synergistic increases in the expression of these two IEGs could have applications in enhancing learning and memory by altering the excitatory/inhibitory balance in the brain and promoting synaptic rearrangement.

Understanding the signaling pathways involved in CEPO and IGF-1 treatment can provide insight into the mechanisms involved in producing the synergistic effect seen during the combination treatment. Previous studies have implicated PI3K/AKT and MAPK signaling pathways downstream of both IGF-1 [43] and CEPO [17,39,40]. Studies have also shown robust increases in phosphorylated AKT after combination EPO and IGF-1 treatments in cerebrocortical cultures [13,15]. In the current study, we show increased phosphorylation of both AKT and ERK after five minutes and one hour of IGF-1 and CEPO + IGF-1 treatment, but not CEPO treatment. In contrast with a previous study which noted a synergistic increase in phosphorylated AKT after combination treatment [13], our study does not show a more robust phosphorylation of either AKT or ERK after combination treatment when compared to the intensity after the individual treatments. The previous study attributed the synergistic effects of EPO and IGF-1 to robust and prolonged phosphorylation of AKT that lasted up to nine hours after stimulation. Our study shows robust, but not synergistic, AKT phosphorylation up to five hours after stimulation, but the intensity of the phosphorylation appears to be decreasing at that time point. As well, despite robust ERK phosphorylation at the five minute and one hour time points, ERK phosphorylation decreases significantly after three and five hours of stimulation. From these data, a prolonged phosphorylation of AKT is noted, which aligns with previous work. A prolonged phosphorylation of ERK is also observed but with a shorter lifetime than that seen with AKT phosphorylation. Despite these robust increases, no synergistic increase in phosphorylation is observed after combination treatment. 

IEG activation is thought to be regulated by multiple pathways including ERK, RhoA-actin, p38 MAPK, and PI3K [44,45,46]. To further implicate specific pathways in mediating the increased IEG expression seen after trophic factor treatment, three kinase inhibitors—GDC-0994, AZD-1480, and MK-2206—were used to inhibit either ERK, JAK2, or AKT, respectively. We found that both ERK and JAK2 inhibition drastically reduced IEG expression while AKT inhibition did little to change the overall pattern of gene expression. These results indicate the importance for ERK and JAK2 signaling in mediating IEG upregulation after CEPO and IGF-1 treatment, while not implicating AKT signaling in this process. Previous research has noted the importance for ERK-mediated regulation of IEG expression [45,47,48,49], which our results would corroborate. Since the IGF1R is a tyrosine kinase receptor, JAK2 signaling would be important for initiating its downstream signaling cascades, which could explain why inhibiting JAK2 causes such a robust decrease in gene expression. One pattern of gene expression that is interesting to note, however, is the expression of *Npas4*, *Inhba*, and *Nptx2*, which show a similar pattern of expression after JAK2 inhibition as seen in the uninhibited samples after three hours of trophic factor stimulation. More research would need to be done to determine what might be mediating this, as it is currently unclear.

The last part of this study looked at the levels of the receptors—EPOR, CD131, IGF1R, and IR—in PC-12 cells to determine whether the receptors were present and, if so, at what levels they were expressed. While all four receptors are expressed, the level at which they are expressed is drastically different. CD131 showed the lowest number of receptors/cell and IGF1R showed the highest, with an expression over 50-fold higher than the other three receptors. Both IR and EPOR showed a similar number of receptors/cell, with EPOR being slightly higher. This is interesting because it could be an underlying reason as to why we see comparatively weak CEPO signaling effects in relation to robust IGF-1 signaling. In future studies, it will be important to study this combination in a cell line showing similar numbers of these receptors to determine whether receptor levels are impacting the pattern of IEG expression we are seeing in this study.

It is also interesting to note whether our trophic factors can induce the expression of receptors of other trophic factors. As CEPO has been suggested to act via a CD131-EPOR heteromer, it is interesting to note that CD131 was not induced by CEPO alone but by CEPO + IGF-1 in combination. The increase in IGF1R after CEPO treatment demonstrates the potential that one trophic factor (in this case CEPO) could induce the expression of a different trophic factor’s receptor (in this case IGF1R, the receptor for IGF-1). This phenomenon could allow one trophic factor to not only act via its own signaling pathway but also increase the signaling capacity for other trophic factors.

The need for newer therapies that improve cognition in neurodegenerative diseases and mood disorders is increasingly apparent. Combination therapies are gaining popularity and could prove to be more efficacious than single compounds. Our gene profile data show that CEPO and IGF-1 can work synergistically to induce IEG’s with known roles in cognition. This finding suggests that (1) CEPO is a promising replacement for EPO in pro-cognitive studies and (2) that a combination approach aimed at improving cognition may be more effective than a single molecule treatment. Future studies should focus on testing this combination in vivo to examine whether combining these trophic factors improves cognitive functioning at the behavioral level.

## Figures and Tables

**Figure 1 life-13-01826-f001:**
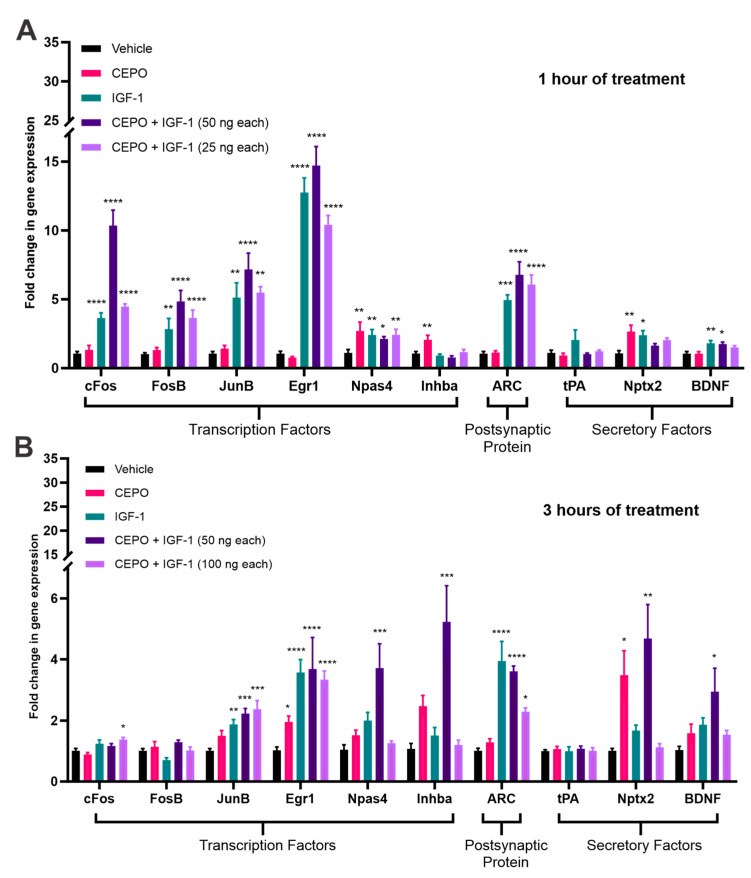
Comparative analysis of trophic factor-induced gene regulation in PC-12 cells. (**A**) PC-12 cells treated for one hour with CEPO (100 ng/mL), IGF-1 (100 ng/mL), CEPO + IGF-1 (50 ng/mL each), or CEPO + IGF-1 (25 ng/mL each) (*n* = 6). (**B**) PC-12 cells treated for three hours with CEPO (100 ng/mL), IGF-1 (100 ng/mL), CEPO + IGF-1 (50 ng/mL each), or CEPO + IGF-1 (100 ng/mL each) (*n* = 5). Quantitative PCR analysis was performed using gene specific primers. Gene regulation is expressed relative to vehicle levels. Error bars are ± SEM. Significance was determined when compared to the vehicle-treated group. * *p* < 0.05, ** *p* < 0.01, *** *p* < 0.001, and **** *p* < 0.0001 one-way ANOVA with Dunnett’s multiple comparisons post hoc test or the Kruskal–Wallis test with Dunn’s multiple comparisons post hoc test.

**Figure 2 life-13-01826-f002:**
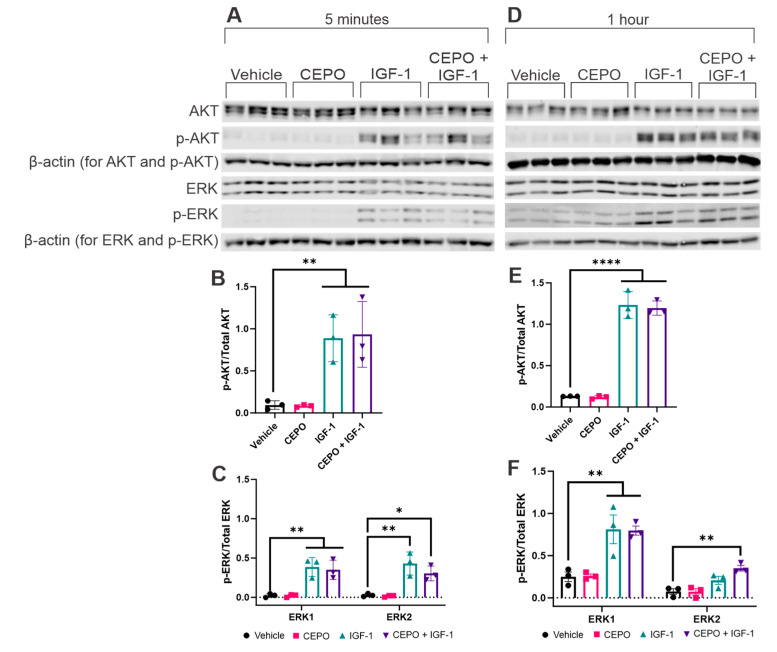
Trophic factor-induced phosphorylation of ERK and AKT in PC-12 cells. Western blot of PC-12 cells treated with CEPO (100 ng/mL), IGF-1 (100 ng/mL), and CEPO + IGF-1 (50 ng/mL each) for (**A**) 5 min or (**D**) 1 h. (**B**,**C**) Quantification of Western blot bands from (**A**) (*n* = 3). (**E**,**F**) Quantification of Western blot bands from (**D**) (*n* = 3). Error bars are ± SD. Significance was determined when compared to the vehicle-treated group. * *p* < 0.05, ** *p* < 0.01, and **** *p* < 0.0001 one-way ANOVA with Dunnett’s multiple comparisons post hoc test.

**Figure 3 life-13-01826-f003:**
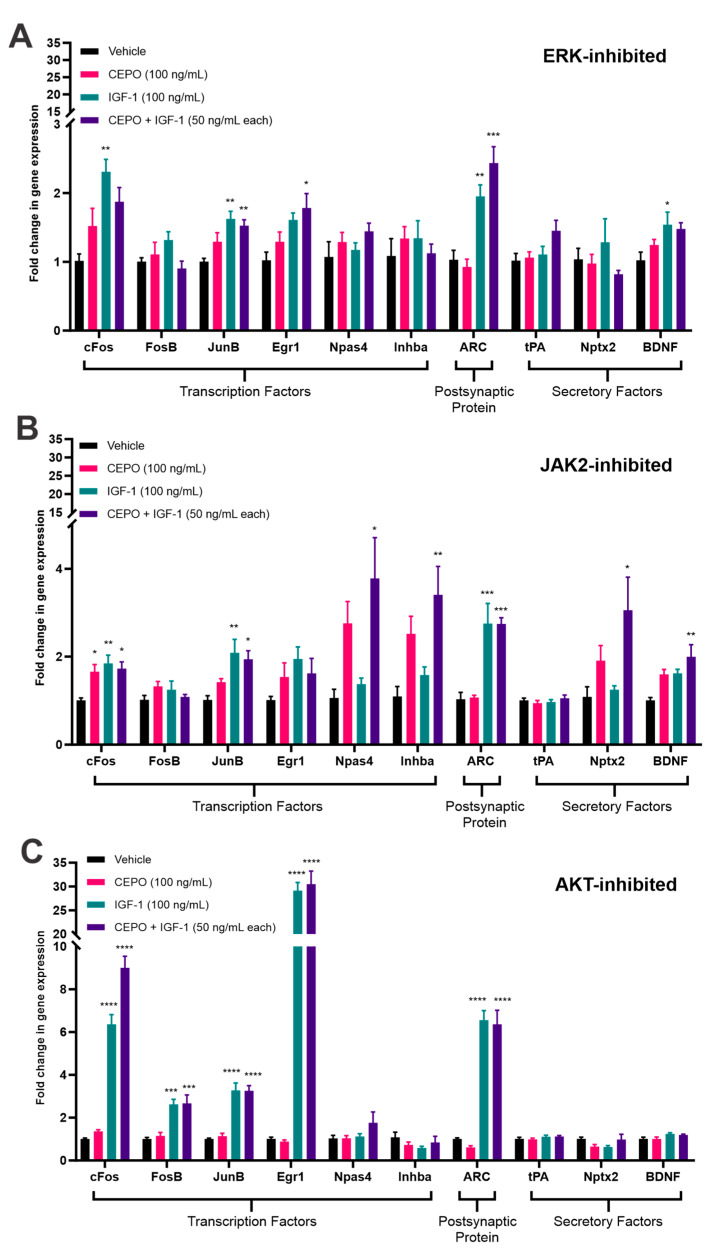
Comparative analysis of trophic factor-induced gene regulation after kinase inhibition in PC-12 cells. Cells were inhibited with (**A**) GDC-0994 (10 µM, 1 h) (*n* = 4/treatment group), (**B**) AZD-1480 (2 µM, 2 h) (N = 4/treatment group), or (**C**) MK-2206 (5 µM, 1 h) (*n* = 5/vehicle-treated group and *n* = 4/other treatment groups) before being stimulated for one hour with either CEPO (100 ng/mL), IGF-1 (100 ng/mL), or CEPO + IGF-1 (50 ng/mL each) in the presence of inhibitor. Quantitative PCR analysis was performed using gene specific primers. Gene regulation is expressed relative to vehicle levels. Error bars are ± SEM. Significance was determined when compared to the vehicle-treated groups. * *p* < 0.05, ** *p* < 0.01, *** *p* < 0.001, and **** *p* < 0.0001 one-way ANOVA with Dunnett’s multiple comparisons post hoc test or the Kruskal–Wallis test with Dunn’s multiple comparisons post hoc test.

**Figure 4 life-13-01826-f004:**
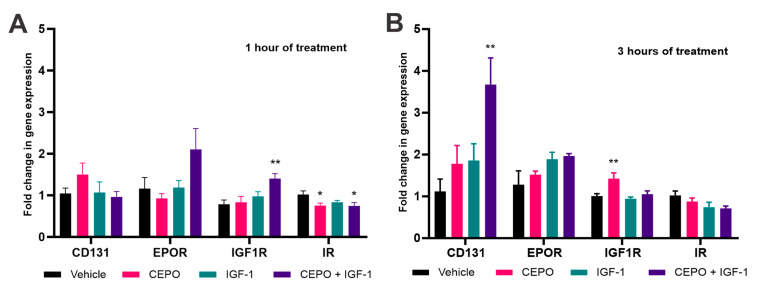
Comparative analysis of trophic factor-induced gene regulation of receptors in PC-12 cells. (**A**) PC-12 cells treated for one hour with CEPO (100 ng/mL), IGF-1 (100 ng/mL), or CEPO + IGF-1 (50 ng/mL each) (*n* = 6). (**B**) PC-12 cells treated for three hours with CEPO (100 ng/mL), IGF-1 (100 ng/mL), or CEPO + IGF-1 (50 ng/mL each) (*n* = 5). Quantitative PCR analysis was performed using gene specific primers. Gene regulation is expressed relative to vehicle levels. Error bars are ± SEM. Significance was determined when compared to the vehicle-treated group. * *p* < 0.05 and ** *p* < 0.01, one-way ANOVA with Dunnett’s multiple comparisons post hoc test or the Kruskal–Wallis test with Dunn’s multiple comparisons post hoc test.

**Figure 5 life-13-01826-f005:**
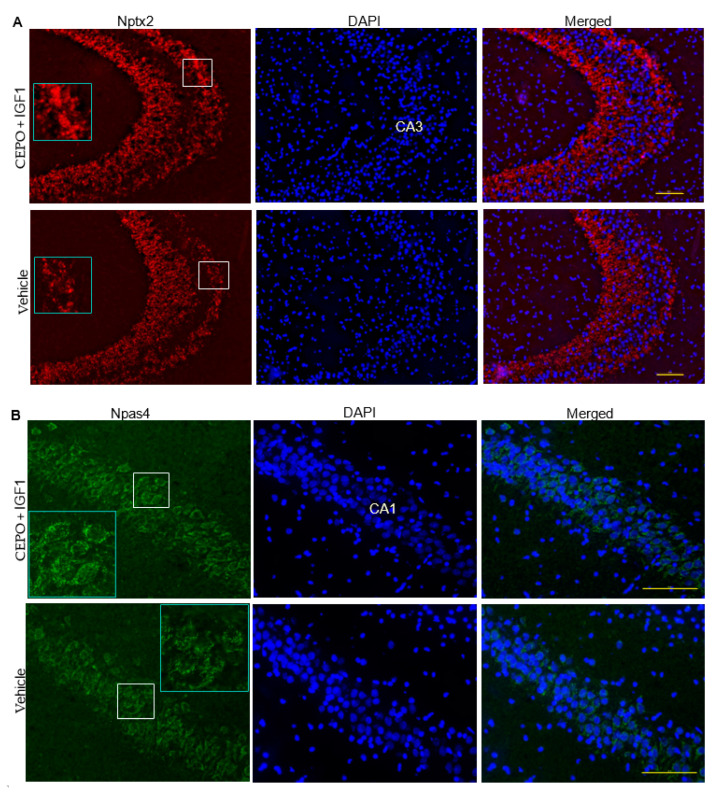
Immunohistochemical analysis of Nptx2 and Npas4 expression. (**A**) Coronal, cryocut rat brain sections from the dorsal hippocampus were immunochemically stained for Nptx2 after four days of CEPO + IGF-1 or vehicle administration. Magnified inset shows localized regions of marked higher expression after trophic factor treatment. Representative images from the CA3 region are shown from *n* = 3. (**B**) Npas4 expression in the CA1 region of the rat dorsal hippocampus after CEPO + IGF-1 or vehicle administration. Magnified inset shows slightly different expression pattern of Npas4 relative to DAPI-stained nuclei after trophic factor treatment. Representative images are shown from *n* = 3. Scale bars = 100 µm.

**Table 1 life-13-01826-t001:** Comparative analysis of receptor levels in PC-12 cells. Digital droplet PCR analysis was performed using gene specific primers. Total copies of target DNA per cell were calculated for each sample with the averages being recorded.

EPOR	CD131	IGF1R	IR
7.0	0.9	584.8	5.2

## Data Availability

The datasets analyzed during the current study are available from the corresponding author on reasonable request.

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
