# Peer review of "Synergism of Carbamoylated Erythropoietin and Insulin-like Growth Factor-1 in Immediate Early Gene Expression"

_life, 2023, doi:10.3390/life13091826_

Round 1

Reviewer 1 Report

The authors hypothesized that CEPO and IGF-1, when used in combination, would synergistically activate immediate early genes. The authors examined IEG expression after treatment with CEPO, IGF-1, or CEPO + IGF-1 in rat pheochromocytoma cells and the protein expression of Npas4 and Nptx2 in the rat brain. The manuscript is well written and structured, the introduction provides sufficient background. The authors obtained interesting results. However, some questions require clarification.

1. The abstract is very large. The section should be a total of about 200 words maximum. Please consider the possibility of reducing.

2. Line 106: “To differentiate the cells into neuronal cells …” lacks a corresponding literature reference.

3. Line 179-181: It seems necessary to indicate the protocol code and date of approval.

4. Line 183: “CEPO (25 µg/kg) + IGF-1 (50 µg/kg)…” It is necessary to justify why the authors used these doses.

5. Line 199: “Elements imaging software. CEPO + IGF-1 and vehicle-treated sections were captured using identical exposure sections.” It seems necessary to indicate: mm from bregma.

6. Line 201-207: Have you checked the data for the normality of the distribution? You have small groups (N = 4 – 6). In cases where the hypothesis on the normal distribution cannot be accepted, it is necessary to use nonparametric tests.

7. Fig. 1B: CEPO + IGF-1 (50 ng/mL each), or CEPO + IGF-1 (100 ng/mL each). How do you explain the less significant result after high-dose combination treatment?

8. Fig. 2. : N values should be added.

9. Line 359-361: “Both Nptx2 and Npas4 showed higher levels of protein expression after CEPO + IGF-1 treatment, albeit in different hippocampal regions.” On the basis of what do the authors make this conclusion? Has a quantitative analysis been carried out?

10. Fig. 5: Please specify “Scale bars = “. Figure would be easier to understand if you added a general outline. As an example: Ref. 35. Ramamoorthi, K.; Fropf, R.; Belfort, G.M.; Fitzmaurice, H.L.; McKinney, R.M.; Neve, R.L.; Otto, T.; Lin, Y. Npas4 Regulates a Transcriptional Program in CA3 Required for Contextual Memory Formation. Science 2011, 334, 1669-1675. Supplemental Figure 3.

11. It is very interesting to investigate whether the combination of these trophic factors improves the cognitive functions of animals.

Author Response

We thank the reviewer for their comments and the points that were raised. Described below in detail are our responses and how we have addressed the various points in the manuscript. New and edited text are highlighted.

  1. The abstract is very large. The section should be a total of about 200 words maximum. Please consider the possibility of reducing.

We thank the reviewer for this comment. The abstract is now shortened to 200 words.

  1. Line 106: “To differentiate the cells into neuronal cells …” lacks a corresponding literature reference.

We have inserted a literature reference to Wiatrak et al. (2020) after this statement. This article does a good job of giving an overview of PC-12 cells and explains why differentiation via NGF is necessary when using this cell line for neuroscience research. This change can be found in line 97.

  1. Line 179-181: It seems necessary to indicate the protocol code and date of approval.

We thank the reviewer for pointing this out. We have added both the protocol code and date of approval to the methods section of this paper, found in lines 175-176.

  1. Line 183: “CEPO (25 µg/kg) + IGF-1 (50 µg/kg)…” It is necessary to justify why the authors used these doses.

We have added a sentence in which we reference previous work where the same, or similar, doses were used to justify why we chose these doses. This change can be found in lines 178-180.

  1. Line 199: “Elements imaging software. CEPO + IGF-1 and vehicle-treated sections were captured using identical exposure sections.” It seems necessary to indicate: mm from bregma.

We thank the reviewer for this comment. We have added the mm from bregma to the materials and methods IHC section. This change can be found on lines 197-198.

  1. Line 201-207: Have you checked the data for the normality of the distribution? You have small groups (N = 4 – 6). In cases where the hypothesis on the normal distribution cannot be accepted, it is necessary to use nonparametric tests.

We thank the reviewer for this comment. We have checked the normality of the distribution for each of the data sets. There were a few cases in which normality was not met and the statistical analyses were updated to nonparametric tests. These changes were updated in both the materials and methods section and in each figure/corresponding results section.

  1. Fig. 1B: CEPO + IGF-1 (50 ng/mL each), or CEPO + IGF-1 (100 ng/mL each). How do you explain the less significant result after high-dose combination treatment?

We appreciate this point from the reviewer. We have updated the results section to include information on why we might be seeing a less significant result after high-dose combination treatment. This addition can be found on lines 244-253. As another general point that we felt need not be added to the paper but can also be used to explain this phenomenon with respect to EPOR, there is a 1000-fold difference in affinity between the high-affinity and low-affinity sites of the EPOR dimers (with these sites being located on opposite receptor monomers). At high ligand concentrations, this difference in affinities could be negated because there is now an over-abundance of ligand present to bind to each receptor site individually. Thus, the general sequence of binding events—high-affinity site first, then low-affinity site—is overruled and we see ligand binding to each monomeric EPOR unit and thus decreasing the ability to signal through this receptor.

  1. Fig. 2. : N values should be added.

Thank you for this comment. The N values for the bar graphs have been added to the figure legend. The revision can be found on lines 294 and 295.

  1. Line 359-361: “Both Nptx2 and Npas4 showed higher levels of protein expression after CEPO + IGF-1 treatment, albeit in different hippocampal regions.” On the basis of what do the authors make this conclusion? Has a quantitative analysis been carried out?

We thank the reviewer for this comment. This experiment was meant to be a semi-quantitative analysis to observe whether our trophic factors caused any expression changes to IEGs at the protein level in the rat brain.

  1. Fig. 5: Please specify “Scale bars = “. Figure would be easier to understand if you added a general outline. As an example: Ref. 35. Ramamoorthi, K.; Fropf, R.; Belfort, G.M.; Fitzmaurice, H.L.; McKinney, R.M.; Neve, R.L.; Otto, T.; Lin, Y. Npas4 Regulates a Transcriptional Program in CA3 Required for Contextual Memory Formation. Science 2011, 334, 1669-1675. Supplemental Figure 3.

We thank the reviewer for this comment. We have added the scale bar length to the figure legend, found in line 392. We have also added a supplementary figure showing an outline of our hippocampal regions of interest (Figure S2), with in-text reference found on lines 376-378.

  1. It is very interesting to investigate whether the combination of these trophic factors improves the cognitive functions of animals.

Thank you for this comment. We plan to investigate this in future experiments.

Reviewer 2 Report

This study is well designed and the manuscript is well and clearly written. No objection to publish it in present form.

Author Response

We thank the reviewer for their comments and the points that were raised. Described below in detail are our responses and how we have addressed the various points in the manuscript. New and edited text are highlighted.

This study is well designed and the manuscript is well and clearly written. No objection to publish it in present form.

We greatly appreciate this comment from the reviewer.

Reviewer 3 Report

The presented research of Rothschadl et al. evaluates the potential synergistic effects of combining erythropoietin (EPO) and insulin-like growth factor-1 (IGF-1) as trophic factors in neuroprotection. While previous studies have shown benefits of combining EPO and IGF-1, the limitations of EPO's impact on hematological parameters prompted the examination of carbamoylated EPO (CEPO), a non-erythropoietic derivative. The study reveals that CEPO and IGF-1 produce a synergistic increase in immediate early gene (IEG) expression in PC-12 cells. Inhibition studies point to ERK and JAK2 pathways as mediators of these effects. Furthermore, protein expression experiments in rat brains support the efficacy of the CEPO and IGF-1 combination in modulating behavior and cognition. These findings highlight the potential value of the synergistic use of trophic factors for neuroprotection.

There are several minor comments regarding the manuscript.

1. Materials and Methods: The corresponding section of the manuscript should include primer sequences for qPCR. Alternatively, a separate file with additional information can be included.

2. PCR conditions were not provided. What were the primer annealing temperature, elongation time, etc.? How many cycles were used in the reaction?

3. Line 203, it is mentioned that "A one-way ANOVA with Dunnett’s multiple comparisons post hoc test was used to look for significance in the qPCR studies when analyzing each gene individually". However, the figure captions state "one-way ANOVA with Bennett’s multiple comparisons post hoc test".

4. Figure 1 and further: Please indicate gene names using standard nomenclature in italics for clarity.

Author Response

We thank the reviewer for their comments and the points that were raised. Described below in detail are our responses and how we have addressed the various points in the manuscript. New and edited text are highlighted.

  1. Materials and Methods: The corresponding section of the manuscript should include primer sequences for qPCR. Alternatively, a separate file with additional information can be included.

We thank the reviewer for this comment. We have now added a supplementary table containing the gene abbreviations, names, corresponding NCBI Ascension number, and forward and reverse primer sequences. Corresponding in-text mentions can be found on lines 138 and 158.

  1. PCR conditions were not provided. What were the primer annealing temperature, elongation time, etc.? How many cycles were used in the reaction?

We thank the reviewer for this comment. The PCR conditions were added to the materials and methods and can be found on lines 139-140.

  1. Line 203, it is mentioned that "A one-way ANOVA with Dunnett’s multiple comparisons post hoc test was used to look for significance in the qPCR studies when analyzing each gene individually". However, the figure captions state "one-way ANOVA with Bennett’s multiple comparisons post hoc test".

Thank you for catching this. Stating “Bennett’s multiple comparisons post hoc test” was a typing error. This has been changed to Dunnett’s. This error was made in the legends for Figures 1, 2, 3, and 4 and the revision can be found on lines 266, 297, 337, and 370, respectively.

  1. Figure 1 and further: Please indicate gene names using standard nomenclature in italics for clarity.

Thank you for this comment. We have indicated the gene names using standard nomenclature during their first mention in the paper. These changes can be found on lines 216-217 and 222-224. 

Round 2

Reviewer 1 Report

The authors provided a detailed answer to all the questions and comments raised. Most of them have led to amendments that, in my opinion, improve, supplement or clarify the quality of their work.